

# Next generation atmosphere-ocean climate modelling for storm surge hazard projections

Cléa Denamiel[1], Ivica Vilibić[1]

[1]Ruđer Bošković Institute, Division for Marine and Environmental Research, Bijenička cesta 54, 10000 Zagreb, Croatia

*Correspondence to*: Cléa Denamiel (cdenami@irb.hr)

**Abstract.** Due to on-going global warming, extreme storm surges are expected to threaten a greater number of coastal communities worldwide. However, global and regional climate simulations of extreme events are still not accurate enough to respond to the growing needs of the local decision makers to prepare for these rising hazards. We present a new approach using (sub-)kilometre-scale coupled atmosphere-ocean-wave models and demonstrate the feasibility to provide meter-scale

assessments of the impact of climate change on storm surge hazards. As a proof of concept, we focus in the Adriatic Sea and analyse the sea levels of two kilometre-scale 31-year long simulations used in evaluation and extreme warming modes. First, we demonstrate that, at 1-km resolution, the model errors are reduced by up to a third compare to state-of-the-art regional and global models. Second, we show that meter-scale storm surge results – obtained by further downscaling extreme events extracted from the kilometre-scale simulations – contrast with the previously published literature. In particular, we found that

some understudied regions of the Adriatic coast might be more vulnerable to sea level rise and atmospherically driven storm surges induced by extreme climate warming than the well-researched Venice Lagoon. Following these preliminary results, we present a newly developed methodology directly downscaling extreme events from global climate models. Within this framework, the numerical resources, previously spent to produce long-term simulations, are used efficiently to quantify the climate change uncertainty and to properly assess the meter-scale storm surge hazards.

## 1 Introduction

In this era of accelerated temperature rise, the climate research community still faces two main challenges. The first is the need to convince the global and local decision makers that human-driven global warming will have a strong societal impact on the sectors of energy, food, agriculture, health, urbanization, environment and could lead to important financial and economic burdens (Creutzig et al., 2022). The second is the critical importance to provide to the same decision makers more

accurate climate projections of the entire Earth system, for them to better adapt to the societal impact of future extreme events like droughts, storms, sea level rise, etc. (Smith et al., 2014).

Importantly, due to projected mean sea level rise (Hamlington et al., 2020) and intensification of atmospheric storms like hurricanes and tropical cyclones (Chen et al., 2020) under climate warming, low-lying populated coastlines are expected to be more and more exposed to coastal hazards, especially extreme storm surges. Consequently, local decision makers should





start working on adaptation plans in order to build new infrastructures at minimal possible cost (Vallejo and Mullan, 2017). However, extreme sea level hazard assessments are highly influenced by local processes shaped by both mesoscale atmospheric processes (e.g., occurrences, durations and intensities of the storms) and local geomorphology of the coastal areas (e.g., coastal and harbour resonances, topographic shoaling), while global and regional climate models are often too coarse to properly represent both extreme storms and endangered coastlines (Hinkel et al., 2021).

However, in the coupled atmosphere-ocean modelling community, the accuracy of the models, and thus of the climate projections, has been closely following different breakthroughs in computational science. This includes availability of more powerful numerical resources, better storage facilities, more efficient programming languages, etc. Historically (Fig. 1), the first coupled atmosphere-ocean global circulation model (AO-GCM) was created in the 1970s by Manabe et al. (1975) and Bryan et al. (1975) to derive – for 100-year long periods with resolutions of about 500 km – the temperature trends resulting
from the global increase of the greenhouse gases. Nowadays the GCM resolutions can reach up to 25-50 km (Iles et al., 2020) and ensembles of models are run to better quantify the uncertainty of the climate projections. Further, in the 2000s-2010s, coupled atmosphere-ocean regional climate models (AO-RCMs) using dynamical downscaling of the GCM results were implemented with higher resolutions and better physics (Giorgi, 2019). They aimed at studying the regional processes and providing vulnerability, impact, and adaptation assessments with about 10 km of accuracy. Additionally, in the 2000s, as
the climate community gained better knowledge of the impact of climate change, the need for kilometre-scale climate models (Schär et al., 2020) better suited to characterize extreme events (with higher resolution and less physical parametrizations) also emerged. But, due to their extreme computational costs, these models were only developed in the atmosphere for simulations (1) ranging from few days during extreme events to 31-year periods, and (2) using the Pseudo-Global Warming (PGW) method for future climate projections. This is only in the 2020s that the PGW method was extended
to the ocean (Denamiel et al., 2020a) and implemented in the first coupled atmosphere-ocean kilometre-scale climate model. Additionally, in order to provide storm surge hazard projections, the kilometre-scale climate results were further downscaled to a sub-kilometre-scale ocean model representing the complex meter-scale geomorphology of the coastal areas (e.g., harbors, bays, etc.). This next generation approach to climate modelling was first implemented and tested for short-term simulations in the semi-enclosed Adriatic basin in the Mediterranean region (Denamiel et al., 2020a, 2020b, 2021a), where
the orographically-shaped bora and sirocco winds better represented at kilometre-scales are driving storm surges in lagoons, bays and harbours only seen at (sub-)kilometre scales (Fig. 2).

Here, we present the storm surge results of the so-called Adriatic Sea and Coast (AdriSC) climate modelling suite (Denamiel et al., 2019) (1) at the kilometre-scale, for two 31-year long simulations (evaluation run for the 1987-2017 period and climate projection run using the PGW method for the far-future 2070-2100 period) and (2) at the (sub)-kilometre-scale, for an
ensemble of moderate to extreme events extracted from the long-term simulations. Additionally, the added value of the presented downscaling strategy is discussed and an alternative methodology to long-term climate simulations is proposed in order to better balance model accuracy and numerical cost. Finally, the local application of the (sub-)kilometre-scale methodology to the Adriatic Sea is only used as a proof of concept and all the presented approaches and results can be



replicated and/or adapted at any location in the world where extreme sea level hazard assessments fully depend on the

accurate representation of the complex geomorphology of the coastal areas.

## 2 Models and Methods

### 2.1 Adriatic Sea and Coast (AdriSC) climate model

The Adriatic Sea and Coast (AdriSC) climate modelling suite (Denamiel et al., 2019) is composed of two different modules which can be used together or independently to quantify a variety of climate-related processes. The AdriSC general

circulation module is designed to run long-term climate simulations. It is based a modified version of the (COAWST; Warner et al., 2010) model and couples online the Weather Research and Forecasting (WRF; Skamarock et al., 2005) model with the Regional Ocean Modelling System (ROMS; Shchepetkin and McWilliams, 2009). In this module, (1) the two atmospheric grids at respectively 15-km and 3-km of resolution are two-way nested, while (2) the one-way nested ocean grids at 3-km and 1-km of resolution are forced by the 3-km atmospheric grid. For the presented climate study, the AdriSC

general circulation module is not coupled to a wave model. However, the AdriSC extreme event module which further downscales the general circulation results for short-term simulations (i.e., 1.5 day), also includes wave modelling. It couples offline the WRF results downscaled at 1.5-km of resolution with the fully coupled unstructured ADvance CIRCulation – Simulating WAves Nearshore (ADCIRC-SWAN; Dietrich et al., 2012) storm surge barotropic model at up to 10 m of resolution along the Adriatic coasts. Detailed descriptions of the modelling suite (e.g., physics setup, tidal- and river-

forcing, coupling, grid and mesh description, etc.) can be found in Denamiel et al. (2018, 2019, 2021b) and Pranić et al. (2021).

The Pseudo-Global Warming (PGW) method is used to run the AdriSC general circulation module. It was created to downscale the sparse results of Regional Climate Models (RCMs) to kilometre-scale simulations and is based on the principle that the impact of climate change can be assessed by imposing an additional climatological change to the forcing

used to produce the evaluation run (Prein et al., 2015; Kendon et al., 2017; Denamiel et al., 2020a, 2020b). In the AdriSC climate model, for the atmosphere, the 6-hourly ERA-Interim (Dee et al., 2011) air temperature, relative humidity and horizontal wind velocities three-dimensional and surface reanalysis results are modified by adding climatologic changes. In the ocean, for the Mediterranean Sea, the daily MEDSEA (Simoncelli et al., 2019) ocean temperature, salinity and currents three-dimensional reanalysis results are also modified by adding climatologic changes. In this study, the PGW method was

applied to the coupled atmosphere-ocean LMDZ4-NEMOMED8 RCM model (Hourdin, F. et al., 2006; Beuvier, J. et al., 2010) forced by the IPSL-CM5A-MR GCM model (simulations r1i1p1) and part of the CORDEX experiment (Giorgi et al., 2009; Giorgi and Gutowski, 2015).

As a proof of concept that coupled atmosphere-ocean kilometre-scale climate modelling could be achieved, two 31-year long simulations have been carried out with the AdriSC general circulation module: (1) an evaluation run for the 1987-2017

period forced by the 6-hourly ERA-Interim and the daily MEDSEA reanalysis products and fully evaluated against an





extensive ocean and atmospheric dataset (Denamiel et al., 2021b; Pranić et al., 2021) and (2) a far future climate projection run for the 2070-2100 period following the Representative Concentration Pathway 8.5 (RCP 8.5) scenario forced with the PGW method presented above. Each of these simulations required 18 months to be completed with a continuous run using 260 CPUs on the European Centre for Middle-range Forecast (ECMWF) supercomputing facilities. Additionally, the 600 TB

of hourly climate data generated by the two 31-year long runs is stored on the ECMWF tape system.

## 2.2 Analysis of the AdriSC ROMS 1-km sea levels

Providing that this study mainly focuses on the impact of climate change on atmospherically driven extreme sea levels and that the ocean RCMs and reanalysis products implemented in the Mediterranean Sea do not account for global sea level rise, the strategy of using detrended sea levels included tides, seiches, storm surges but not sea level rise is adopted as follows.

First, the AdriSC ROMS 1-km hourly results are extracted from the multiple files of the 31-year long simulations and merged into a single file. Then, the Theil-Sen trend estimation method (Gilbert, 1987) – insensitive to outliers and significantly more accurate than simple linear regression for skewed and heteroskedastic data – is used to detrend the merged 31-year long AdriSC ROMS 1-km hourly sea level results.

However, despite not being taken into account in the AdriSC kilometre-scale climate simulations, global sea level rise

cannot be ignored in extreme sea level hazard studies. Consequently, the results from the RCP 8.5 scenario of the IPCC-AR5 2015 ensemble (Church et al., 2013) of 20 global models – provided at 1° resolution and including 10 geophysical sources that drive long-term changes in sea levels (e.g., Antarctic dynamic ice and surface mass balance, global thermostatic anomaly, terrestrial water, glacial isostatic adjustment, etc.) – are added to the detrended AdriSC ROMS 1-km sea levels. It should be noted that only 12 models of the AR5 ensemble provide results in the Mediterranean Sea. Practically, the IPCC-

AR5 2015 yearly results are interpolated in space and time to generate hourly results covering the AdriSC ROMS 1-km grid for both the 1987-2017 and the 2070-2100 periods.

Finally, the 10-year, 30-year and 50-year return periods are extracted from the AdriSC detrended 1-km hourly sea levels during the 1987-2017 period following the extreme value theory that derives the Generalized Extreme Value (GEV) distribution (Embrechts et al., 1997; Kotz and Nadarajah, 2000). Practically, the hourly sea levels are fitted to the GEV

distribution at each point of the AdriSC ROMS 1-km grid using the block maxima method for annual sea level maxima. Then, from these fitted distributions it is estimated how often the extreme quantiles occur with a certain return level (in this study 10, 30 and 50 years). These spatially varying return period values are then used to define three categories of atmospherically driven events: (1) moderate events for sea levels between the 10-year and 30-year return periods, (2) severe events for sea levels between the 30-year and 50-year return periods and (3) extreme events for sea levels above the 50-year

return period. It should be noted that even so-called moderate events have the potential to cause material damages and human casualties (e.g., flood, drowning, etc.).





### 2.3 Adriatic storm surges and coastal hazard assessments

Storm surges are derived at each point of 6 different sub-domains (Venice and Marano Lagoon, Gulf of Trieste, Rijeka, Split and Mali Ston bays). The number of points within the 6 sub-domains for both the AdriSC ROMS 1-km grid and the AdriSC

ADCIRC-SWAN unstructured mesh are (1) 454 and 718 points for the Venice Lagoon, (2) 121 and 316 points for the Marano Lagoon, (3) 497 and 792 points for the Gulf of Trieste, (4) 558 and 6694 points for the Rijeka Bay, (5) 665 and 15596 points for the Split Bay and (6) 105 and 10749 points for the Mali Ston Bay. To obtain robust statistics, storm surge hazard in each sub-domain is defined when at least one point of the sub-domain falls within the three categories defined by the spatially varying return periods. All sub-domain points considered, the number of unique days with moderate, severe and

extreme events derived from the AdriSC ROMS 1-km detrended hourly sea levels with the return-period method described above are: (1) 38 for the 1987-2017 period and (2) 37 for the 2070-2100 period under RCP 8.5 scenario. With SLR added to the AdriSC ROMS 1-km detrended hourly sea levels, the number of unique days rises to: (1) 70 for the 1987-2017 period and (2) 6229 for the 2070-2100 period under RCP 8.5 scenario. It should be noted that the AdriSC modelling suite has been principally developed to study climate change along the till now under researched eastern Adriatic coast, and that many

bathymetric data remain confidential along the Italian coast and principally in the Venice and Marano lagoons. Consequently, the ADCIRC-SWAN mesh resolution in these lagoons might not allow for proper adaption plans but the presented methodology can still be used to downscale the AdriSC ROMS 1-km sea level results with the specialized models at higher resolution developed by the Venice research community (e.g., Umgiesser et al., 2004).

As this study mostly focused on atmospherically driven sea levels and as running about 6300 events would have been too

costly in terms of numerical resources, the coastal hazards in the 6 sub-domains are derived with the AdriSC extreme event module, for 1.5-day long simulations, but only for the unique events selected from the detrended hourly sea levels (i.e., 38 for the 1987-2017 period and 37 for the 2070-2100 period). However, SLR is added to the ROMS 1-km sea level results used to force the ADCIRC-SWAN model in order to account for the non-linear interactions between waves, tides, mean sea levels, seiches, and atmospherically driven surges. The coastal hazard assessments are then extracted from the maximum

wind speed and associated direction, sea levels, significant wave height and peak period as well as for the minimum pressure, for each sub-domain and for all moderate to severe storms derived for each sub-domain point in order to produce robust statistics. In this analysis, for each subdomain and each variable, the number of baseline conditions represents 100 % of the occurrences in order to visualize the impact of climate change on the number of RCP 8.5 conditions. Consequently, due to the wet and dry set-up used in the ocean model, percentages of RCP 8.5 conditions might slightly vary between

atmospheric and ocean results.



## 3 Results

### 3.1 Evaluation of the AdriSC ROMS 1-km sea levels and the storm surge extraction

The AdriSC detrended 1-km hourly sea levels are first evaluated against 11 tide-gauges stations along the Adriatic Sea for the 1987-2017 period (Fig. 2 and Table 1). The sea level data is extracted from tide gauges located: (1) in the Venice Lagoon

(at Punta della Salute) and the Gulf of Trieste maintained by the Italian Institute of Marine Science (ISMAR) of the National Research Council (CNR), (2) along the western Italian coast at Ravenna, Ancona, Ortona and Vieste maintained by the Italian National Institute for Environmental Protection and Research (ISPRA), and, finally, (3) along the Croatian coast at Rovinj, Zadar, Split, Ploče and Dubrovnik maintained by the Hydrographic Institute of the Republic of Croatia (HHI). However, it should be noted that: (1) the measurements in the Venice Lagoon are available with sampling periods varying

from 1 to 6 hours, (2) coverage is about 2 times higher from the Italian than the Croatian measurements (Table 1), (3) only the Trieste station achieves a 100 % coverage for the 1987-2017 period and (4) the measurements from the ISPRA tide gauges are only publicly available from 1999. The basic statistical analysis presented in Table 2 shows that correlation between observations and model are equal or above 0.80 for all the stations (except 0.798 at Ploče) and root mean square error is below 15 cm (except 15.5 cm at Venice). Further differences between the percentiles derived from observations and

model are always below 10 cm and are on average around 5 cm. However, differences in maximum value can reach up to 26 cm in Venice. Finally, the comparison done for the measured and modelled detrended hourly sea level distributions above the 95th, 99th, 99.9th percentiles (Fig. 3) shows that: (1) for most stations the AdriSC ROMS 1-km model is capable to reproduce the most extreme atmospherically driven sea level events, (2) some outlier values were not removed from the measurements at Ravenna station which resulted in an unrealistic distribution of the sea-levels above the 99.9th percentile,

(3) the strongest difference in distributions occur at the Zadar and Ploče stations with the lowest hourly coverages (25% and 13% respectively; Table 1) for the sea-levels above the 99.9th percentile and (4) for the Trieste station with 100% of hourly coverage (Table 1), the model distributions nearly perfectly fit the observational ones. Consequently, it is found that the observed storm surges (i.e., detrended tide-gauge data above 95th, 99th and 99.9th percentiles) are well reproduced by the AdriSC ROMS 1-km model results.

To test the reliability of the method to properly select the moderate to extreme storm surges, the events occurring only in the Venice Lagoon sub-domain are extracted from the 70 ones including SLR during the 1987-2017 period, and compared to the Venice flood events selected by Lionello et al. (2021) in Table 3. With the method 20 unique events are extracted from the AdriSC ROMS 1-km detrended sea levels with added SLR while only 10 flood events above 1.4 m were recorded in Venice. From the 20 extracted events from the AdriSC ROMS 1-km sea levels, (1) 7 are moderate events which may have sea level

values lower than 1.4 m at Venice (labelled as Too low/Non-existent), (2) 4 are extreme events but clearly not recorded at Venice (labelled as Non-existent), (3) 3 slightly missed the real event (shift of 1 to 4 days), and (4) 5 are perfectly matching with the recorded events at Venice. However, only two events recorded at Venice are missed. These differences can be explained as follows: (1) events for the Venice Lagoon are selected when only 1 point of the sub-domain is falling into the 3



storm surge categories while observations are made in Venice, (2) moderate events can generate sea levels below 1.4 m at
Venice, (3) synchronization of the maximum tidal signal and the maximum atmospherically driven extreme sea levels may
be missed by the AdriSC model if the modelled storm is slightly shifted in time, (4) SLR estimate may be slightly different
from the measured evolution of the mean sea level at Venice which, for example, also accounts for subduction and (5) the 1-
km resolution of the AdriSC ROMS model might be too coarse to properly reproduce the floods in Venice influenced by the
local bathymetry (e.g., Ferrarin et al., 2021). Despite these discrepancies, the method is found to show skills in extracting the
most extreme sea level events in Venice which is the only location in the Adriatic Sea where long-term flood recordings
exist.

## 3.2 Impact of climate change on the spatial sea level distributions

We first present the spatial variations of the detrended 1-km sea level maximum and skewness results (Figs. 4 and 5) for the
baseline conditions (top panels) and the climate adjustments defined as the difference between RCP 8.5 and baseline
conditions (bottom panels). Special attention is given to three different areas with complex geomorphology: (1) the northern
Adriatic including the Venice and Marano lagoons as well as the Gulf of Trieste, (2) the Kvarner Bay where hurricane-
strength easterly bora winds are blowing over a complex network of islands and (3) the region of the Dalmatian Islands
including the Split and Mali Ston bays.

In the shallow northern part of the Adriatic Sea, the baseline maximum results (Fig. 4) perfectly capture the well-known
increase in sea level extremes (above 1.7 m in the Veneta Lagoon). These events are driven by non-linear interactions
between tides, atmospheric surges, wave setup and basin fundamental seiches (Marcos et al., 2009) which can only be
simulated with high-resolution limited-area atmosphere-ocean models (Cavaleri et al., 2010). In particular, the AdriSC
model reproduces sea level increases (1) of up to 30 cm between the western and eastern regions of the Venice Lagoon, (2)
of 10-15 cm within the Marano Lagoon and (3) of 15-20 cm in the northernmost part of the Gulf of Trieste. The associated
baseline skewness (Fig. 5), reaching a maximum of 0.1 in the Veneta Lagoon and being quasi-null in the Gulf of Trieste, is
the smallest of the entire Adriatic basin. This is due to the predominance of tides which have the highest amplitudes (Malačič
et al., 2000) in this region of the Adriatic and thus account for a large part of the sea level distributions. Further, the
skewness remains positive due to the impact of the southerly sirocco winds which produce the most extreme storm surges in
the northern Adriatic (Bertotti et al., 2011). However, as the Gulf of Trieste is also influenced by the easterly bora winds
decreasing the coastal sea levels, the skewness becomes quasi-null there. Additionally, the variation of skewness within the
Venice Lagoon reflects the high sensitivity of sea level distributions to the local forcing and thus the importance to
downscale climate models to the kilometre-scale for storm surge assessments. Concerning the climate adjustments, for the
maximum sea levels an increase of 10-15 cm is projected within the western Venice and Marano lagoons and the Gulf of
Trieste, while in general a decrease of 5-10 cm is seen within the northern Adriatic shelf except along the Po River plume
(Fig. 4). Further, in the Veneta Lagoon, the maximum sea levels are expected to largely decrease by 15-20 cm. These
features, only be seen at the kilometre-scale, reveal a probable shift in the wind directions producing the most extreme sea





levels under extreme warming. The associated skewness (Fig. 5) is expected to overall uniformly increase by 25-30 % under RCP 8.5 conditions and the sea level distributions will tend to have more pronounced right tails. Consequently, the extreme sea level events will tend to be more intense under RCP 8.5 conditions, despite the decrease of maximum sea levels at certain locations.

In the Kvarner region, the baseline maximum detrended sea levels can reach up to 1.1 m in the Rijeka Bay (Fig. 4), but do not seem to be strongly influenced by the complex network of islands. The associated positive skewness (Fig. 5) is quite large (0.25-0.30 in the north to 0.50 in the south) due to the combined influence of severe offshore bora winds (Grisogono and Belušić, 2009) and an amphidromic point (Malačić et al., 2000) modulating the lowest semidiurnal tidal components in the middle Adriatic. Further, climate change is expected (1) to lower in the north and increase in the south the maximum sea levels by up to 10-15 cm and (2) to increase the skewness and thus the occurrence of the extreme sea levels by up to 30 %. Consequently, this region seems more affected by the general changes within the basin – i.e., maximum sea level and skewness increase in the middle Adriatic – than by the orographically-shaped local processes.

However, the Dalmatian Islands region perfectly illustrates the importance of capturing the complex coastal geomorphology in climate models. Indeed, the baseline maximum sea levels reaching up to 0.75 m within the small bays and along the islands of the domain (Fig. 4), are clearly driven by the local seiches and topographic amplification of the incoming offshore long ocean-waves (Rabinovich, 2009). As for the Kvarner Bay, the associated skewness (Fig. 5), still influenced by the geomorphology of the region, is largely driven by the modulation of the tidal amplitudes in the Adriatic Sea. Further, extreme warming in this region is projected to have the strongest impact on extreme sea levels in the whole Adriatic basin: maximum sea levels are expected to increase by up to 15-20 cm over the entire Split Bay and by up to 10-15 cm in the Mali Ston Bay, with an increase of 20 % in skewness and thus in the number of extreme events.

**3.3 Impact of climate change on the alongshore sea level distributions**

To further quantify the impact of climate change at the kilometre-scale, we present the spatial variations of the extreme sea level distribution statistics (95th, 99th, 99.9th percentiles, maximum, skewness and kurtosis) along the 3600 km of the Adriatic coastline (anticlockwise from east to west, Fig. 6) for both baseline conditions (top panels) and climate adjustments under RCP 8.5 conditions (bottom panels).

As already seen in the previous analyses, baseline 95th to 99.9th sea level percentiles and maximums increase towards the northern Adriatic, but localized peaks also occur along the entire coastline within bays and lagoons (e.g., Dürres, Varano Lagoon). These peaks are generally more pronounced for maximum sea levels than for sea level percentiles, indicating that the severity of extreme events is larger in regions capable to locally amplify the extremes. Further, both skewness and kurtosis have the largest values in the middle Adriatic where the tidal amplitudes are the lowest and the tails of the sea level distributions are only influenced by the atmospherically driven extreme sea levels. In contrast, in the northern Adriatic between the Gulf of Trieste and the Po River delta, the skewness is quasi-null, and the kurtosis is negative, indicating the strongest influence of the tidal signal despite the presence of extreme events.





However, along the Venice Lagoon and the Po River delta, the kurtosis presents an important negative peak while the
        skewness remains slightly positive, which indicates that the sea level extremes are occurring as outliers. These results are in
        good agreement with the fact that the largest sea level extremes in the Mediterranean are known to occur along the Po River
        sedimentary plain (including the Venice Lagoon) which is extremely vulnerable to subsidence (Teatini et al., 2011).
        Concerning the impact of climate change under RCP 8.5 scenario, if the maximum sea levels are expected to increase by 10-
20 cm in the bays and lagoons along the Adriatic coastline as seen previously, it is more interesting to see that the 95th to
        99.9th percentile values are nearly not affected (i.e., changes of the order of few cm) by extreme warming. Additionally, the
        skewness and kurtosis are generally expected to increase along the entire coastline, except at locations where the maximum
        sea levels decrease strongly (e.g., up to 40 cm in the Varano Lagoon) and thus the kurtosis also decreases. Consequently,
        these results suggest that extreme storm surges may be more frequent and more destructive but that, in average,
atmospherically driven sea levels will not be as strong as in the present day. Finally, the substantial decrease in maximum
        sea levels in Dürres or Varano Lagoon associated with a decrease of kurtosis may indicate a shift in direction of the extreme
        winds capable to produce extreme sea levels under RCP 8.5 scenario.

### 3.4 Impact of climate change on storm surge hazards

        We now use classical engineering methods to project the impact of climate change on the duration and frequency of
moderate, severe and extreme events (Fig. 7). The storm surge hazards are derived from both the AdriSC 1-km detrended
        hourly sea levels only (top panels) and with sea level rise (SLR) added (bottom panels). The analysis is presented for six sub-
        domains – the Venice and Marano lagoons, Gulf of Trieste, Rijeka, Split and Mali Ston bays – where the impact of climate
        change was found to be the strongest.
        For the storm surge only, the analysis of the unique occurrences of moderate to extreme events, highlights the large spatial
variability of the climate change impact under RCP 8.5 scenario (Fig. 7, top left panel). For example, the occurrences of
        moderate to extreme conditions are expected to decrease by 2 hours in the Venice Lagoon but to be multiplied by more than
        2.5 in the Marano Lagoon located less than 90 km further in the northern Adriatic. Also, the severe and extreme conditions
        are expected to increase by approximately 1.5 times in the Venice Lagoon, 7 times in the Marano Lagoon, 2.5 times in the
        Split Bay and 3 times in the Mali Ston Bay but to decrease by 3 times in the Rijeka Bay. The analysis of the occurrences of
moderate to extreme events averaged over the entire sub-domains (Fig. 7, top right panel) confirms that extreme climate
        change conditions will increase severe and extreme events for all sub-domains – even in the Gulf of Trieste where severe
        events, non-existent in the baseline conditions, will occur. The exception is the Rijeka Bay, where on average only moderate
        events will take place. Additionally, under these far future conditions, areas like the Marano Lagoon and the Split and Mali
        Ston bays may face more extreme storm surge conditions for which they are not prepared nowadays.
When SLR is added, the number of unique days with moderate to extreme events, including all sub-domain points, is
        multiplied by nearly 2 and more than 150 for the baseline and RCP 8.5 conditions, respectively. Due to this dramatic change
        in mean sea level (i.e., up to 0.5 m SLR under RCP 8.5 scenario), the occurrences of the moderate events are expected to



increase by 1 to 3 orders of magnitude for all the sub-domains (Fig. 7, bottom panels). Severe and extreme storm surge conditions are expected to occur in average: (1) more than 2500 hours, instead of 2-5 hours under the baseline conditions, in 290 Split and Mali Ston bays, (2) 150 hours instead of 1.5 hours in the Rijeka Bay, and (3) between 20 and 40 hours, instead of less than 10 hours, in the Venice and Marano lagoons and in the Gulf of Trieste. Consequently, independently of the intensification of the atmospherically driven storm surges, the Rijeka, Split and Mali Ston bays are found to be the locations the most endangered by SLR in the Adriatic Sea.

**3.5 Sub-kilometre-scale coastal hazard assessments**

In order to quantify the storm surge hazards, we run the AdriSC extreme event module for the ensemble of unique days extracted from the storm surge only analysis (Fig. 7, top left panel). However, we also add SLR to the AdriSC 1-km detrended sea levels forcing the unstructured ocean mesh for these events. Here, we present the distributions of maximum sea levels, significant wave height, peak period, wind speed and associated direction as well as minimum pressure for all the 6 sub-domain points, under both baseline and RCP 8.5 conditions for the selected moderate to extreme daily events (Fig. 8).

As seen in previous analyses, under extreme warming, the occurrences of moderate to extreme conditions are expected to rise by 80 % in the Marano Lagoon, 20 % in the Gulf of Trieste and about 10 % in the Split and Mali Ston bays. However, a decrease of about 15 % in the Rijeka Bay and 30 % in the Venice Lagoon is also simulated. These spatial variations of the storm surge hazards can be explained by the changes in atmospheric conditions. In fact, under RCP 8.5 conditions, the intensification of the maximum wind speeds by up to 5 m/s in the Split and Mali Ston sub-domains is accompanied by a 305 strong shift in direction (absence of westerly directions, above 180 °N) and a slight increase in the minimum pressure (about 2-3 hPa). In the Venice Lagoon the direction of the maximum winds stays similar, but the minimum pressure drops (absence of values above 990 hPa). This drop of minimum pressure is also seen in the Marano Lagoon and, to a smaller extent, in the Gulf of Trieste, and is associated with a small shift of direction (increase of north-easterly directions below 90 °N). In the Rijeka Bay, the wind speeds are decreasing (up to 3 m/s), and the associated directions are strongly shifting (absence of 310 westerly directions above 180 °N) while the minimum pressure is slightly dropping (2-3 hPa). Consequently, a shift of the low-pressure system driving the southerly sirocco events responsible for the strongest northern Adriatic storm surges may be expected under extreme warming. Due to this shift of the atmospheric conditions under RCP 8.5 scenario, storm surges are expected (1) to reach up to 2.4-2.7 m in the Split and Mali Ston bays (instead of 0.9-1.1 m in the baseline conditions), (2) to increase by 20 to 40 % for values above 2.5 m in the Marano Lagoon and the Gulf of Trieste and (3) to always be below 2.5 315 m in the Venice Lagoon. Concerning the wave hazards under extreme warming, the maximum significant wave heights are expected to be similar to the baseline conditions for all sub-domains, except the Marano Lagoon where they will rise by up to 0.5 m. The associated maximum peak periods are however expected to slightly decrease except in the Marano Lagoon.



## 4 Discussion and conclusions

On the one hand, despite many well-funded public and private projects studying climate change in the Venice Lagoon, the
preliminary results presented in this study reveal that, in the Adriatic Sea, other locations less studied by lack of funding,
interest, expertise, etc., may be more endangered by the direct impact of climate warming. These areas are the Marano
Lagoon where the largest marina of Italy is located, the Split Bay, the second most populated area of Croatia known for its
UNESCO heritage sites (Riemann et al., 2018) and heavily relying on tourism and cruises, and the Mali Ston Bay
internationally known for the quality of its oysters. Scientists, engineers and local decision makers should thus shift their
attention to these locations in order, for example, to better understand the impact of (1) the wave height increase on the
Marano mooring complex, (2) SLR, likely to flood historical towns in the Split area and (3) the storm surge intensification
on the production of oysters in the Mali Ston Bay.

On the other hand, this study is just the first step towards storm surge assessments under climate change, as the uncertainties
linked to climate change must be properly quantified in order to provide meaningful results to the local decision makers.
This is achieved by running ensembles of simulations forced by multiple global climate models under multiple warming
scenarios (Semenov and Stratonovitch, 2010). However, the presented strategy running coupled atmosphere-ocean
kilometre-scale baroclinic models for 31-year long periods is far too prohibitive, in terms of numerical cost, to be used to
create large ensembles of long-term climate simulations.

Further, the presented results based on (sub-)kilometre-scale atmosphere-ocean models seem to contrast with the previously
published literature, which used much coarser climate storm surge models or were focused on specific sites, like the city of
Venice. For example, Vousdoukas et al. (2017) found that, under RCP 8.5 scenario, the 100-year return period of the storm
surges should decrease along the entire Adriatic coastline by 2100. Additionally, Denamiel et al. (2020a) used the AdriSC
modelling suite and the PGW method on an ensemble of 14 extreme events and found that extreme sea levels are expected to
decrease by more than 0.25 m (up to 0.35 m in the Venice Lagoon) over the entire northern Adriatic domain for both RCP
4.5 and RCP 8.5 scenarios. Consequently, the added value of the presented modelling strategy as well as the envisioned
method to account for climate uncertainty at the (sub-)kilometre-scale are discussed hereafter.

### 4.1 Added value of (sub)-kilometre-scale atmosphere-ocean climate modelling for coastal hazard assessments

The most recent extreme sea levels and coastal flooding information on which the European Environment Agency
based the European adaptation to climate change (https://www.eea.europa.eu/ims/extreme-sea levels-and-coastal-flooding)
are mostly derived from the modelling studies of Vousdoukas et al. (2016, 2017). They use the Delft3D model (Deltares,
2014) with 0.2°-0.25° (about 25-31 km) resolution in the Mediterranean Sea and forced by ERA-Interim in evaluation mode
and by CMIP5 for the climate scenarios (Knutti and Sedlacek, 2013). The evaluation of the model done in Vousdoukas et al.
(2016) for the monthly maxima during the 2008-2014 period shows that the RMSEs (Fig. 9, top panel) are about 0.3 m in
Venice and Trieste and 0.12-0.15 m along the western Adriatic coast while they only reach 0.11 m and 0.07-0.11 m,





respectively, for the AdriSC ROMS 1-km results. Further, the comparison of the scatter plots in Venice, highlights that the AdriSC ROMS 1-km hourly results are far less scattered between -0.5 and 0.5 m and better follow the reference line than the Vousdoukas et al. (2016) results (Fig. 9). Finally, the modelling approach of Vousdoukas et al. (2016, 2017) does not address the non-linear interactions between SLR, tidal flows, waves, and storm surges which play an important role for extreme sea levels at the local scale (Le Bars, 2018; Arns et al., 2015; Du et al., 2018; Zijl et al., 2013; Roland et al., 2013).

By contrast, these non-linear effects are fully resolved, at little numerical cost, in the AdriSC extreme event module only running for the selected ensemble of moderate, severe and extreme storm surge events along the Adriatic coast.

The study of Muis et al. (2020) is also relevant to discuss the impact of the resolution on the accuracy of the storm surge results at the climate scale. They compare two versions of a global storm surge model: (1) the GTSMv2.0 model used in the global study of Vousdoukas et al. (2018) with resolutions varying from 5 km along the coasts to 50 km in the deep ocean

forced 6-hourly by ERA-Interim at about 80 km resolution and (2) the GTSMv3.0 model with resolutions varying from 2.5 km along the coasts (1.25 km in Europe) and 25 km in the deep ocean forced hourly by ERA5 at about 31 km resolution. They found that when the results of the two models are evaluated against the Global Extreme Sea Level Analysis (GESLA; Haigh et al., 2022) tide gauge stations for the 1979-2014 period, correlation coefficients and biases are respectively increased at 90 % and reduced at 95 % of the locations including along the northern and western Adriatic coasts. Consequently, the

spatial and temporal resolutions of the atmospheric forcing as well as the spatial resolution of the ocean models play a crucial role in capturing the sea level extremes at the climate scale and the (sub-)kilometre-scale modelling strategy presented in this study is likely to greatly improve the extreme storm surge hazard projections under climate change along the worldwide coastlines.

## 4.2 Towards more efficient (sub)-kilometre-scale simulations accounting for the climate uncertainty

As the added value of (sub)-kilometre-scale approach for storm surge climate modelling has been proven in this study, we now envision to produce robust storm surge hazard assessments by directly downscaling extreme events from global climate model (GCM) ensembles to short-term kilometre-scale and sub-kilometre simulations based on modelling suites similar to the AdriSC model.

In practice (Fig. 10), we propose to use the latest CMIP6 dataset (Eyring et al., 2016) including between 14 and 16 members

providing 6-hourly pressure level results at 100 km resolution for both Shared Socioeconomic Pathways (SSP) 2-4.5 and SSP5-8.5 scenarios. As the spatial and temporal resolution of these models is similar to the ERA-Interim forcing, no change is envisioned concerning the number of nested grids needed to downscale the selected GCMs in order to represent the coastal atmosphere-ocean dynamics. For example, in the Adriatic, we could keep (1) at the kilometre-scale, the 15-km and 3-km resolution nested grids in the atmosphere and the 3-km and 1-km nested grids in the ocean, as well as, (2) at the (sub-

)kilometre-scale, the 1.5-km grid in the atmosphere and the up to 10 m mesh in the ocean. Additionally, within this new framework, wind-wave modelling and SLR can already be added to the kilometre-scale simulations. Finally, with 14-16





ensemble members for at least two SSP and several SLR scenarios, the climate uncertainty is adequately described and propagated from the GCMs to the meter-scale storm surge hazards.

The selection of the extreme sea level events from the GCMs is key to the method. It is based on the generation of targeted
synoptic indices capable to link the synoptic conditions in the atmosphere to the local extreme sea levels recorded at the coastal stations. Such indices have been previously successfully built for detecting convective storms (Chan et al., 2018; Gómez-Navarro et al., 2019) and extreme sea levels driven by meteotsunamis (i.e., tsunami-like waves driven by atmospheric disturbances; Zemunik et al., 2022). For storm surge hazards, these synoptic indices should optimize the extraction of true extreme sea level events from the GCMs and minimize the flagging of false positive. Once the extreme
events selected (including false positives), the GCM results are first downscaled with days- to a week- long kilometre-scale simulations relying on the cascade of nested grids from 15-km to 1-km resolution. However, it should be noted that, for semi-enclosed basins like the Mediterranean Sea, the 100-km resolution of the CMIP6 GCMs might be too coarse to properly resolve the general circulation in the ocean. As storm surges are mainly driven by the atmospheric forcing that can be downscaled following the above methodology, it can be envisioned to use alternative approaches in the ocean (e.g., some
modified version of the PGW method). Finally, for the kilometre-scale simulations with extreme event realizations (i.e., excluding false positive), the results are further downscaled with 1 to 3 day-long sub-kilometre simulations and the targeted storm surge hazard assessments for future projections can be derived.

To conclude, the presented study has demonstrated the feasibility and the added value of producing meter-scale storm surge hazard projections in the Adriatic Sea. Further, we argue that targeted climate modelling can be generalized by directly and
automatically downscaling extreme events from GCM ensembles to short-term (sub-)kilometre-scale simulations. Within this framework we thus propose to use the numerical resources, previously spent to produce regional long-term simulations, to quantify the climate change uncertainty and to properly assess the meter-scale storm surge hazards along the worldwide coastlines with the aim to increase the preparedness of coastal communities to the upcoming rise in sea levels and extreme events.

**Code availability**

The code of the COAWST model as well as the ecFlow pre-processing scripts and the input data needed to re-run he AdriSC climate model in evaluation mode for the 1987–2017 period can be obtained under the OSF FAIR data repository at https://osf.io/zb3cm (last accessed 29/10/2022; Denamiel, 2021).

**Data availability**

The model results as well as the post-processing scripts used to produce this article can be obtained under the Open Science Framework (OSF) FAIR data repository at https://osf.io/2hgfm (last accessed 29/10/2022; Denamiel, 2022).



**Author Contribution**

The study conception and design was done by CD. Material preparation was done by CD. Set-up of the model and simulation were performed by CD. Production of the figures was performed by CD. Analysis of the results was performed 415 by IV and CD. The first draft of the manuscript was written by CD and all authors commented on previous versions of the manuscript. All authors read and approved the final manuscript.

**Competing interests**

The authors declare that they have no conflict of interest.

**Acknowledgments**

We are grateful to the European Centre for Middle-range Weather Forecast (ECMWF) which provided staff support – we particularly thank Xavier Abellan and Carsten Maass – as well as the computing and archive facilities used in this research through the Special Projects "Numerical modelling of the Adriatic–Ionian decadal and interannual oscillations: from realistic simulations to process-oriented experiments" and "Using stochastic surrogate methods for advancing towards reliable meteotsunami early warning systems". We also want to thank (1) the Integrated Climate Data Center (ICDC), CEN of the 425 University of Hamburg, Germany which provided the regional sea level data from IPCC AR5 in netCDF format, (2) the CNR Institute of Marine Science (ISMAR), National Institute for Environmental Protection and Research (ISPRA) and the Hydrographic Institute of the Republic of Croatia (HHI) which provided sea level data for the study, (3) Laurent Li from the Université Pierre et Marie Curie who provided regional climate model ocean–atmosphere results (LMDZ4-NEMOMED8) used in the AdriSC RCP 8.5 climate projection 2070-2100, and (4) Danijel Belušić from the University of Zagreb with who 430 the authors had fruitful discussions on climate modelling of extreme events.

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



## Tables and Figures

**Table 1. Long-term tide-gauge observations available along the western and eastern Adriatic Sea during the 1987-2017 period.**

| | Stations | Location | Period | Sampling | Coverage* |
|---|---|---|---|---|---|
| Western Adriatic coast | Venice | 45.431° N 12.336° E | 1987-2016 | 1 to 6 h | 49 % |
| | Ravenna | 44.492° N 12.283° E | 1999-2017 | 1 h | 56 % |
| | Ancona | 43.625° N 13.507° E | 1999-2017 | 1 h | 61 % |
| | Ortona | 42.356° N 14.415° E | 1999-2017 | 1 h | 60 % |
| | Vieste | 41.888° N 16.177° E | 1999-2017 | 1 h | 59 % |
| Eastern Adriatic coast | Trieste | 45.654° N 13.756° E | 1987-2017 | 1 h | 100 % |
| | Rovinj | 45.083° N 13.628° E | 1987-2017 | 1 h | 33 % |
| | Zadar | 44.123° N 15.235° E | 1991-2017 | 1 h | 25 % |
| | Split | 43.507° N 16.442° E | 1987-2017 | 1 h | 33 % |
| | Ploče | 43.010° N 17.391° E | 2006-2017 | 1 h | 13 % |
| | Dubrovnik | 42.658° N 18.063° E | 1987-2017 | 1 h | 33 % |

*percentage of the AdriSC 1-km hourly results during the 1987-2017 period covered by the observations


**Table 2. Statistics of the comparison between the AdriSC ROMS 1-km detrended sea level model results and the tide-gauge observations during the 1987-2017 period. RMSE and %tile stand for Root Mean Square Error and percentile respectively.**

| | Stations | RMSE | Correlation | 95th %tile* | 99th %tile* | 99.9th %tile* | Maximum* | Range* |
|---|---|---|---|---|---|---|---|---|
| Western coast | Venice | 0.155 | 0.872 | 0.035 | 0.056 | 0.098 | 0.264 | 0.167 |
| | Ravenna | 0.124 | 0.860 | -0.022 | -0.011 | -0.085 | -0.218 | -0.388 |
| | Ancona | 0.094 | 0.847 | -0.025 | -0.034 | -0.046 | -0.144 | -0.399 |
| | Ortona | 0.085 | 0.830 | -0.024 | -0.035 | -0.041 | -0.164 | -0.332 |
| | Vieste | 0.080 | 0.834 | -0.016 | -0.021 | -0.023 | -0.133 | -0.165 |
| Eastern coast | Trieste | 0.146 | 0.878 | -0.026 | -0.015 | 0.001 | 0.025 | -0.091 |
| | Rovinj | 0.119 | 0.877 | -0.030 | -0.026 | -0.013 | 0.127 | -0.063 |
| | Zadar | 0.103 | 0.791 | -0.022 | -0.020 | -0.031 | -0.095 | -0.324 |
| | Split | 0.089 | 0.822 | -0.017 | -0.018 | -0.027 | -0.081 | -0.155 |
| | Ploče | 0.096 | 0.798 | -0.019 | -0.026 | -0.021 | -0.029 | -0.130 |
| | Dubrovnik | 0.081 | 0.833 | -0.006 | -0.006 | -0.011 | 0.030 | -0.037 |

*difference between results obtained with the AdriSC ROMS 1-km model and the tide-gauge observation



**Table 3. Comparison of the AdriSC storm surge events selected by the presented hazard assessment method in the Venice Lagoon**
**during the 1987-2017 period with the long-term records of flooding at Venice (Lionello et al., 2021).**

| Venice flooding events | | | |
|---|---|---|---|
| AdriSC storm surge + SLR | Category | Recorded extreme events above 1.4 m | Recorded water-height (m) |
| 1987-01-14 | Moderate | **Too low/Non-existent** | |
| 1990-12-10 | Moderate | **Too low/Non-existent** | |
| 1992-12-08 | Severe | 1992-12-08 | 1.42 |
| 1992-12-09 | Moderate | **Too low/Non-existent** | |
| 1993-01-03 | Moderate | **Too low/Non-existent** | |
| 2000-11-06 | Moderate | 2000-11-06 | 1.44 |
| **Missed** | | 2002-11-16 | 1.47 |
| 2005-09-18 | Extreme | **Non-existent** | |
| 2005-12-03 | Extreme | **Non-existent** | |
| 2008-11-28 | Moderate | 2008-12-01 | 1.56 |
| 2008-12-11 | Moderate | **Too low/Non-existent** | |
| 2008-12-12 | Extreme | **Non-existent** | |
| 2009-12-19 | Moderate | 2009-12-23 | 1.43 |
| 2009-12-25 | Severe | 2009-12-25 | 1.45 |
| 2010-03-10 | Extreme | **Non-existent** | |
| 2010-12-23 | Extreme | 2010-12-24 | 1.44 |
| 2010-12-24 | Extreme | | |
| 2012-10-28 | Extreme | 2012-11-01 | 1.43 |
| **Missed** | | 2012-11-11 | 1.49 |
| 2013-02-11 | Extreme | 2013-02-11 | 1.43 |
| 2014-01-31 | Moderate | **Too low/Non-existent** | |
| 2015-02-05 | Moderate | **Too low/Non-existent** | |



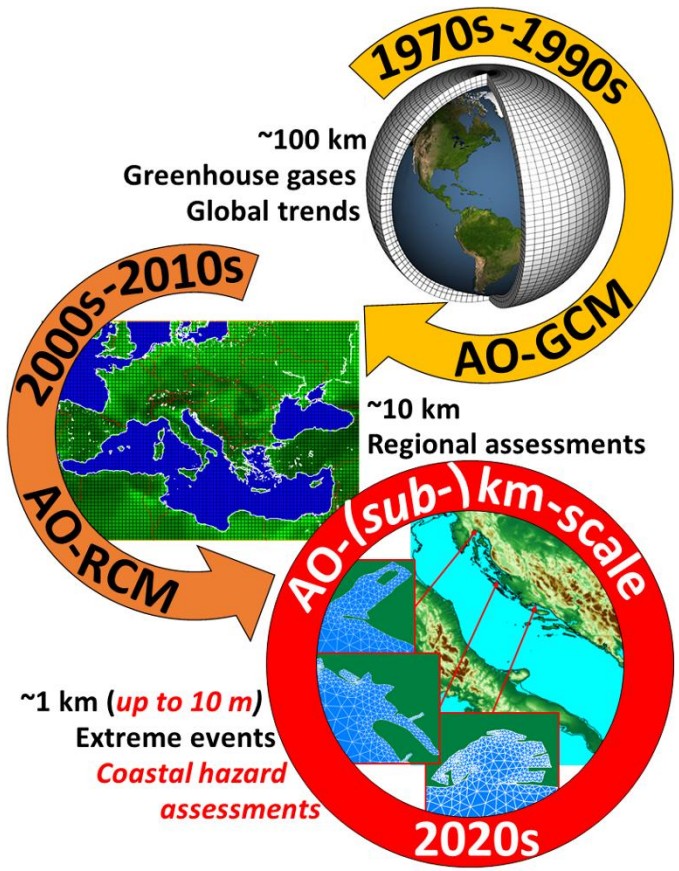

**Figure 1. Historical evolution of the coupled atmosphere-ocean (AO) climate models: from the Global Circulation Models (GCMs) in the 1970s-1990s, to the Regional Circulation Models (RCMs) in the 2000s-2010s, to the recent development (2020s) of (sub-)kilometre-scale models providing extreme sea level hazard assessments at up to 10 m resolution along the coast and within harbours.**



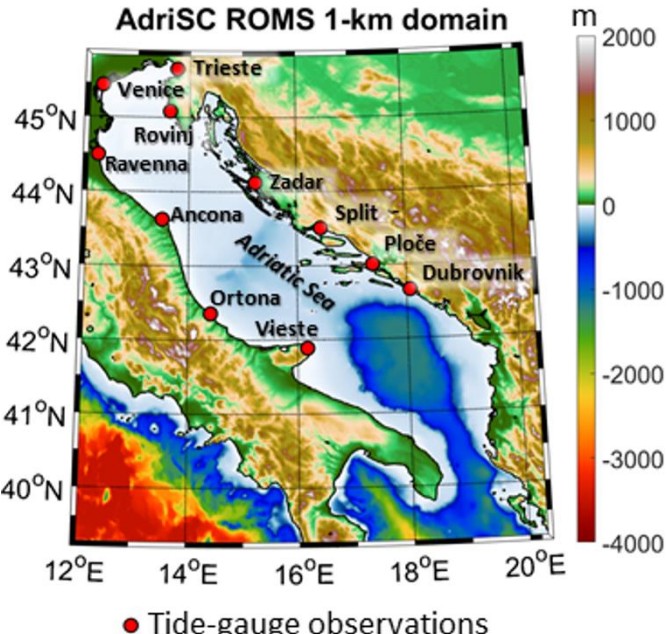

**Figure 2. Topography and bathymetry of the AdriSC ROMS 1-km model domain with the locations of the 11 tide gauges used for the evaluation during the 1987-2017 period.**





**Figure 3. Comparison of the tide gauge observation and the AdriSC ROMS 1-km model distributions of extreme detrended hourly sea levels above the 95th, 99th and 99.9th percentiles at 11 stations along the Adriatic Sea for the 1987-2017 period.**



**Figure 4. Detrended 1-km sea level maximum baseline conditions for the 1987-2017 period (top panels) and RCP 8.5 climate adjustments for the 2070-2100 period (bottom panels) over the entire Adriatic Sea and the northern Adriatic, Dalmatian Island and Kvarner Bay sub-domains.**





**Figure 5. Detrended 1-km sea level skewness baseline conditions for the 1987-2017 period (top panels) and RCP 8.5 climate adjustments for the 2070-2100 period (bottom panels) over the entire Adriatic Sea and the northern Adriatic, Dalmatian Islands and Kvarner Bay sub-domains.**



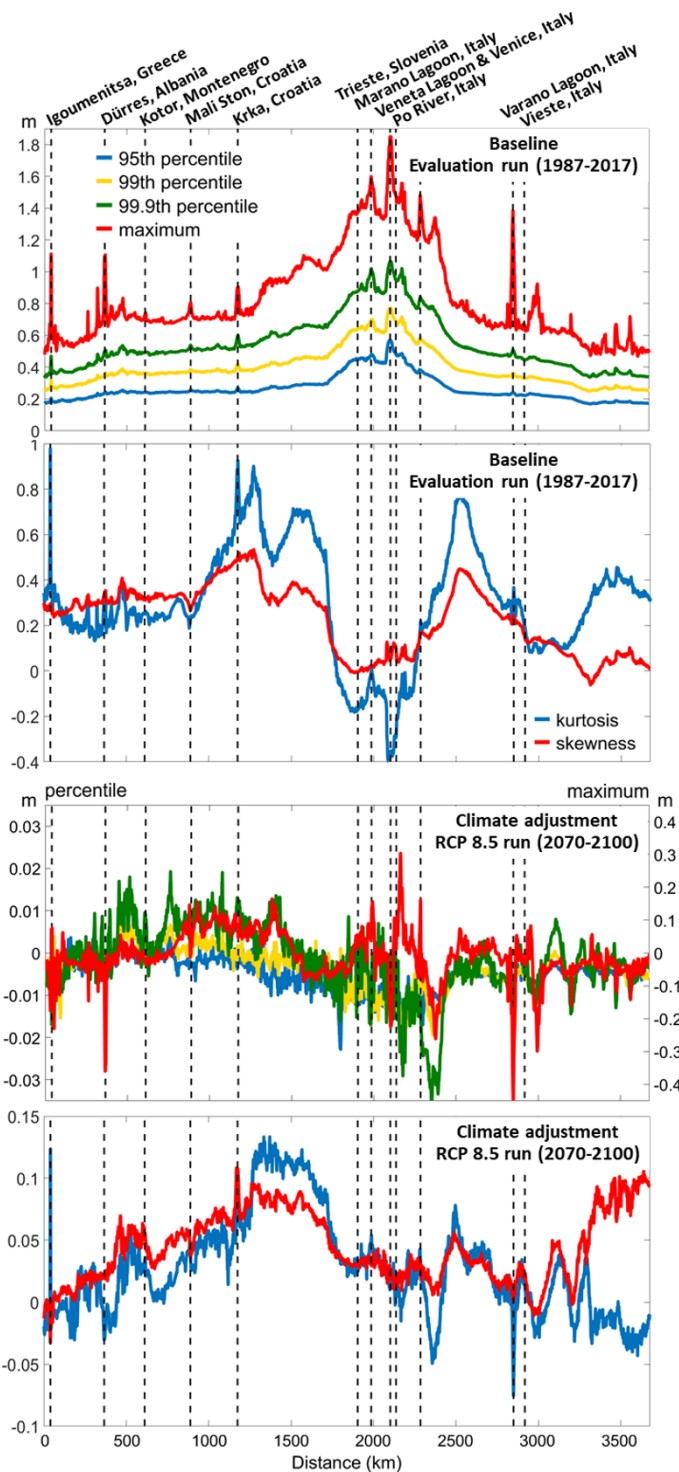

**Figure 6. Detrended 1-km sea level 95th, 99th, 99.9th percentiles, maximum, skewness and kurtosis baseline conditions for the 1987-2017 period (top panels) and RCP 8.5 climate adjustments for the 2070-2100 period (bottom panels) along the Adriatic coastline.**

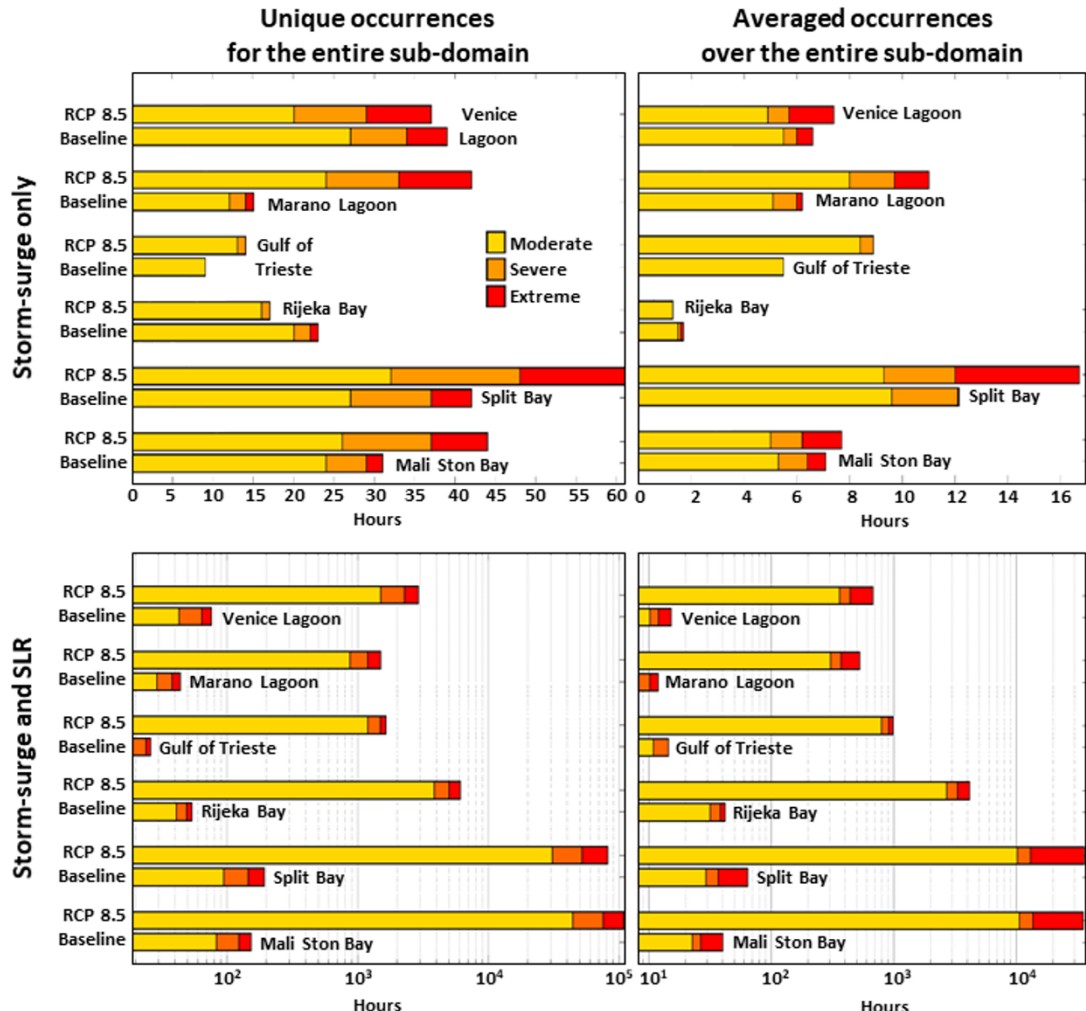

**Figure 7. Storm surge hazard assessments for six Adriatic Sea sub-domains (i.e., Venice Lagoon, Marano Lagoon, Gulf of Trieste, Rijeka Bay, Split Bay and Mali Ston Bay) for three categories: moderate, severe and extreme conditions. The conditions are defined for detrended 1-km sea level values only (top panels) and with additional estimated Sea level Rise (SLR, bottom panels) between the 10- to 30- year return periods (moderate), between the 30- and 50- return periods (severe) and above or equal to the 50-year return period (extreme). All return periods are derived from the baseline storm conditions for the 1987-2017 period. The chosen hazard assessments integrate both events and length of the events by considering the number of hours falling to each category either as a unique occurrence (left panels) or as the average over the number of sub-domain points (right panels).**







**Figure 8. Distributions of the (sub-)kilometer-scale maximum (max.) wind speed and associated direction at 10 m, minimum (min.) mean sea level atmospheric pressure, maximum sea levels, maximum significant wave height and peak period for the baseline and RCP 8.5 moderate, severe and extreme unique daily events derived in Figure 4 storm surge only results but including SLR.**



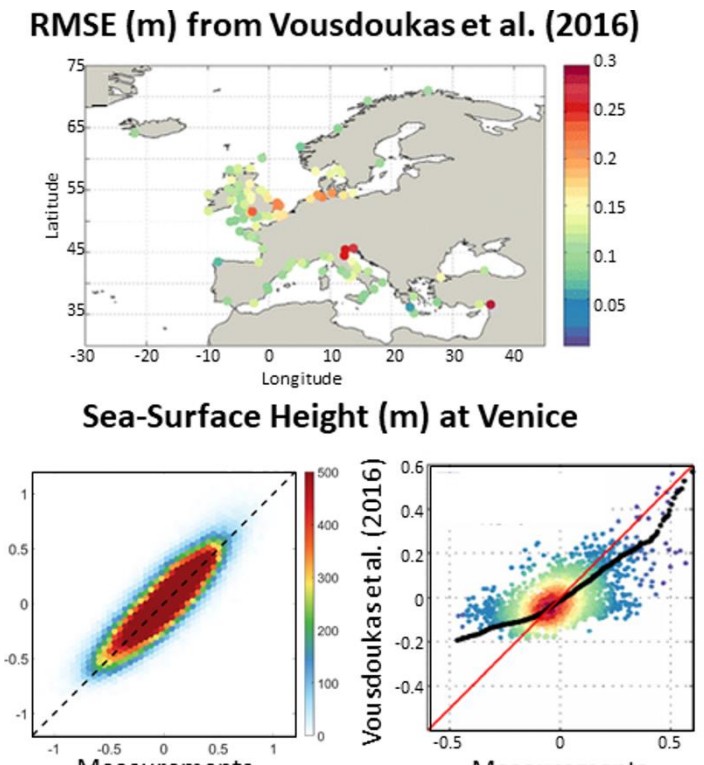

**Figure 9. Comparison of the AdriSC ROMS 1-km and Vousdoukas et al. (2016) model skills in terms of Root Mean Square Error (RMSE; top panel) and scatter plots at Venice (bottom panels) of the sea levels.**

620



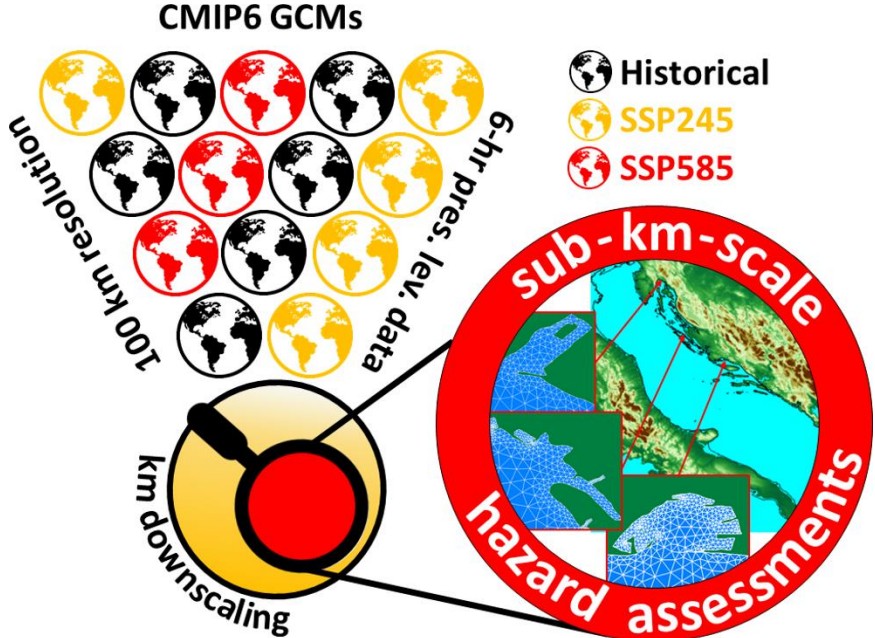

**Figure 10. Direct downscaling of extreme storm surge events from the CMIP6 Global Climate Model (GCM) ensemble of 6-hourly pressure level (pres. lev.) results at 100 km resolution for historical, SSP245 and SSP585 scenarios. First the storm surge events are detected from the GCMs with synoptic indices capable to correlate synoptic conditions and local extreme sea levels. Then kilometer downscaling is applied to all selected events for 3-day long simulations. Finally, sub-kilometer downscaling is applied for 1.5-day long simulations if storm surges are captured at the kilometer-scale and hazard assessments accounting for the climate uncertainty are provided.**

625