# Peer review of "Next generation atmosphere-ocean climate modelling for storm surge hazard projections"

_EGUsphere, 2023_

## Referee Comment (RC1)

**Next generation atmosphere-ocean climate modelling for storm surge hazard projections by *Cléa Denamiel, Ivica Vilibić**

**Comprehensive summary,**

This research aims at developing a new modelling approach for investigating future extreme surges in the Adriatic sea. The methodology combines a continuous run using a regional model to hindcast 31yrs of past climate, with a local model set-up at different locations, and used to simulate several extreme events selected with the annual maxima method on the regional simulation output, the time span of each extreme event simulations is fixed to 1.5 days. Two simulations, a historical reference one and a future one, are used to investigate the impact of future climate on extreme surge. The historical one is set-up using ERAinterim forcing conditions. In contrast, the future one is applying the pseudo-global warming method on the output of one future model (LMDZ4-NEMOMED8 RCM) from the Cordex experiment. Additionally, Sea Level Rise is superimposed on the future simulations with output from the IPCC-AR5-RCP8.5 ensemble in the Mediterranean Sea. Results focus on contrasting historical and future simulations to understand changes in spatial total sea level and surge distributions, from both the regional and local models. Results from the one future climate scenario assessed here, show that several areas along the Adriatic sea coastline are sensitive to future changes in extreme surge.

**Decision**

The topic of the manuscript and the modelling effort put in this research are of great relevance. However, I recommend against publication of the manuscript in its current state. I appreciate the effort put by the two authors, and the potential of their analysis, so I would like to challenge them to resubmit a new version including a specific research aim, sound scientific description of the hypothesis, experimental design, and findings, and how these represent a substantial contribution to understanding future surge hazards in the Adriatic Sea or for modelling of extreme sea levels. I admit that with such large and detailed analysis it can get very complicated, but the current presentation quality is rather low. Additionally, although the scientific significance and quality of this manuscript for the understanding of future surge hazards in the Adriatic sea has the potential to be high, in the current stage is uncertain due to missing details or discussion.

Although I wrote several major and minor comments on the current manuscript, I would like to first share some general insights below to support authors with the resubmission of their manuscript. Please understand them to be constructive.

- The introduction is wide and general. There is little focus on the case study. I also miss insights on what the current scientific gaps are and what are those taken in this manuscript, what is new? is that increase on resolution? The extreme events simulations? why is the PGW used over other methods?

- There is little reflection on what others have done regarding waves and surge projections in the introduction.

- The manuscript is difficult to read, in my opinion using too much decoration, I advise to adopt a more scientific approach on statements. In fact, the positioning is not clear, is a methodological finding or a case study presented?

For example, Line 62- *Finally, the local application of the (sub-)kilometre-scale methodology to the Adriatic Sea is only used as a proof of concept and all the presented approaches and results can be replicated and/or adapted at any location in the world where extreme sea level hazard assessments fully depend on the accurate representation of the complex geomorphology of the coastal areas*

I advise in such a case to present the experiment that validates the hypothesis that upscaling of the methodology is possible *to any location in the world*. More precisely, I would encourage the authors to be cautious with limitations, once their methodology might struggle to simulate future meteo-tsunamis, that are of *great relevance for extreme sea level hazard assessment* in the Adriatic sea.

- The manuscript requires that methods are reproducible, important elements are missing in the methodology. For example, several models such as ROMS, ADCIRC, and SWAN are used, where parameterizations, spatial, and temporal discretization are really impacting results. Details on how the pseudo-global warming method is applied are neither provided.

- Regarding the above, if the manuscript aims at developing a new modelling approach for investigating future extreme surges, I encourage to present the experimental design elaborating on sensitivities or uncertainties arising from model choices.

- Then, how do sensitivities from modelling choice compare with the changes seen in the future scenario simulated? If authors have addressed this in other research publications it is worth to incorporate explicitly in this manuscript.

- The manuscript structure is ok regarding sections used; however I found significant information out of their corresponding section; a few examples are:
  Line 134-138 In methods but it is part of the results section.
  Line 138-143 In methods but it is part of the discussion section.
  Line 188-196 In methods but it is part of discussion
  In general, in the results section new data is described.

- In my opinion, there are choices that are inconsistent, and that are not discussed appropriately. For example:
  If interaction between mean sea level rise and other sea level components are of interest in this research, why the regional model does not account for it, but local ones do? I consider that superimposing mean sea level rise in the boundary conditions of the local models could lead into numerical artifacts in the local results, to accommodate for inconsistent boundary conditions, I would say that, if it is one of the research objectives of this manuscript, at least understanding the effect of this superposition deserves investigation.

- I find figure 1 too general, a proper schematic of the manuscript workflow would help very much on understanding the methodological approach and how it benefits the assessment of future extreme surge.

- In general, consider the use of appendixes when needed.

---

## Referee Comment (RC2)

[referee-annotated manuscript omitted]